# MINILM: Deep Self-Attention Distillation for Task-Agnostic Compression of Pre-Trained Transformers

**Wenhui Wang**    **Furu Wei**[*]    **Li Dong**    **Hangbo Bao**    **Nan Yang**    **Ming Zhou**

Microsoft Research

{wenwan,fuwei,lidong1,t-habao,nanya,mingzhou}@microsoft.com

## Abstract

Pre-trained language models (e.g., BERT [12] and its variants) have achieved remarkable success in varieties of NLP tasks. However, these models usually consist of hundreds of millions of parameters which brings challenges for fine-tuning and online serving in real-life applications due to latency and capacity constraints. In this work, we present a simple and effective approach to compress large Transformer [42] based pre-trained models, termed as deep self-attention distillation. The small model (student) is trained by deeply mimicking the self-attention module, which plays a vital role in Transformer networks, of the large model (teacher). Specifically, we propose distilling the self-attention module of the last Transformer layer of the teacher, which is effective and flexible for the student. Furthermore, we introduce the scaled dot-product between values in the self-attention module as the new deep self-attention knowledge, in addition to the attention distributions (i.e., the scaled dot-product of queries and keys) that have been used in existing works. Moreover, we show that introducing a teacher assistant [26] also helps the distillation of large pre-trained Transformer models. Experimental results demonstrate that our monolingual model[2] outperforms state-of-the-art baselines in different parameter size of student models. In particular, it retains more than $99\%$ accuracy on SQuAD 2.0 and several GLUE benchmark tasks using $50\%$ of the Transformer parameters and computations of the teacher model. We also obtain competitive results in applying deep self-attention distillation to multilingual pre-trained models.

## 1 Introduction

Language model (LM) pre-training has achieved remarkable success for various natural language processing tasks [28, 18, 29, 12, 14, 48, 21, 25]. The pre-trained LMs, such as BERT [12] and its variants, learn contextualized representations by predicting words given their context using large scale text corpora, and can be fine-tuned with additional task-specific layers to adapt to downstream tasks. However, these models usually contain hundreds of millions of parameters which brings challenges for fine-tuning and online serving in real-life applications for latency and capacity constraints.

Knowledge distillation [17, 32] (KD) has been proven to be a promising way to compress a large model (called the teacher model) into a small model (called the student model), which uses much fewer parameters and computations while achieving competitive results on downstream tasks. There have been some works that task-specifically distill pre-trained large LMs into small models [39, 41, 37, 1, 47, 27]. They first fine-tune the pre-trained LMs on specific tasks and then perform distillation.

---

[*] Contact person.

[2]The code and models will be publicly available at `https://aka.ms/minilm`.

Task-specific distillation is effective, but fine-tuning large pre-trained models is still costly, especially for large datasets. Different from task-specific distillation, task-agnostic LM distillation mimics the behavior of the original pre-trained LMs and the student model can be directly fine-tuned on downstream tasks [40, 35, 20, 38].

Previous works use soft target probabilities for masked language modeling predictions or intermediate representations of the teacher LM to guide the training of the task-agnostic student. DistilBERT [35] employs a soft-label distillation loss and a cosine embedding loss, and initializes the student from the teacher by taking one layer out of two. But each Transformer layer of the student is required to have the same architecture as its teacher. TinyBERT [20] and MOBILEBERT [38] utilize more fine-grained knowledge, including hidden states and self-attention distributions of Transformer networks, and transfer this knowledge to the student model layer-to-layer. To perform layer-to-layer distillation, TinyBERT adopts a uniform function to determine the mapping between the teacher and student layers, and uses an additional parameter matrix to linearly transform student hidden states into the same size as its teacher. MOBILEBERT assumes the teacher and student have the same number of layers and introduces the bottleneck module to keep their hidden size the same.

In this work, we propose the deep self-attention distillation framework for task-agnostic Transformer based LM distillation. The key idea is to deeply mimic the self-attention modules which are the fundamentally important components in the Transformer based teacher and student models. Specifically, we propose distilling the self-attention module of the last Transformer layer of the teacher model. Compared with previous approaches, using knowledge of the last Transformer layer rather than performing layer-to-layer knowledge distillation alleviates the difficulties in layer mapping between the teacher and student models, and the layer number of our student model can be more flexible. Furthermore, we introduce the scaled dot-product between values in the self-attention module as the new deep self-attention knowledge, in addition to the attention distributions (i.e., the scaled dot-product of queries and keys) that have been used in existing works. Using scaled dot-product between self-attention values also converts teacher and student representations of different dimensions into relation matrices with the same dimensions without introducing additional parameters to transform student representations. It allows arbitrary hidden dimensions for the student model. Finally, we show that introducing a teacher assistant [26] helps the distillation of large pre-trained Transformer based models.

We conduct extensive experiments on downstream NLP tasks. Experimental results demonstrate that our monolingual model distilled from $\text{BERT}_{\text{BASE}}$ outperforms state-of-the-art baselines in different parameter size of student models. Specifically, the 6-layer model of 768 hidden dimensions distilled from $\text{BERT}_{\text{BASE}}$ is $2.0\times$ faster, while retaining more than $99\%$ accuracy on SQuAD 2.0 and several GLUE benchmark tasks. Moreover, our multilingual model distilled from $\text{XLM-R}_{\text{BASE}}$ also achieves competitive performance with much fewer Transformer parameters.

## 2 Preliminary

Multi-layer Transformers [42] have been the most widely-used network structures in state-of-the-art pre-trained models. In this section, we present a brief introduction to the Transformer networks and the self-attention mechanism, which is the core component of the Transformer. We also present the existing approaches on knowledge distillation for Transformer networks, particularly in the context of distilling a large Transformer based pre-trained model into a small Transformer model.

### 2.1 Backbone Network: Transformer

Given a sequence of input tokens, the vector representations ($\{\mathbf{x}_i\}_{i=1}^{|x|}$) are computed via summing the corresponding token embedding, position and segment embedding. Transformer [42] is used to encode contextual information for input tokens. The input vectors $\{\mathbf{x}_i\}_{i=1}^{|x|}$ are packed together into $\mathbf{H}^0 = [\mathbf{x}_1, \cdots, \mathbf{x}_{|x|}]$. Then stacked Transformer blocks compute the encoding vectors as:

$$\mathbf{H}^l = \text{Transformer}_l(\mathbf{H}^{l-1}), \ l \in [1, L]$$

where $L$ is the number of Transformer layers, and the final output is $\mathbf{H}^L = [\mathbf{h}_1^L, \cdots, \mathbf{h}_{|x|}^L]$. The hidden vector $\mathbf{h}_i^L$ is used as the contextualized representation of $\mathbf{x}_i$. Each Transformer layer consists of a self-attention sub-layer and a fully connected feed-forward network. Residual connection [16] is employed around each of the two sub-layers, followed by layer normalization [2].

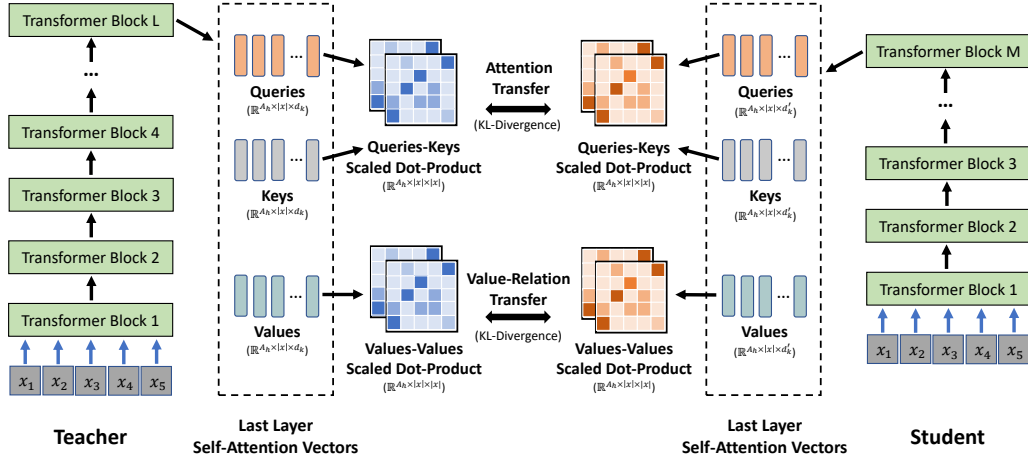

Figure 1: Overview of Deep Self-Attention Distillation. The student is trained by deeply mimicking the self-attention behavior of the last Transformer layer of the teacher. In addition to the self-attention distributions, we introduce the self-attention value-relation transfer to help the student achieve a deeper mimicry. Our student models are named as MINILM.

**Self-Attention**    In each layer, Transformer uses multiple self-attention heads to aggregate the output vectors of the previous layer. For the $l$-th Transformer layer, the output of a self-attention head $\mathbf{O}_{l,a}$, $a \in [1, A_h]$ is computed via:

$$\mathbf{Q}_{l,a} = \mathbf{H}^{l-1}\mathbf{W}_{l,a}^Q, \ \mathbf{K}_{l,a} = \mathbf{H}^{l-1}\mathbf{W}_{l,a}^K, \ \mathbf{V}_{l,a} = \mathbf{H}^{l-1}\mathbf{W}_{l,a}^V$$

$$\mathbf{A}_{l,a} = \operatorname{softmax}(\frac{\mathbf{Q}_{l,a}\mathbf{K}_{l,a}^\mathsf{T}}{\sqrt{d_k}})$$

$$\mathbf{O}_{l,a} = \mathbf{A}_{l,a}\mathbf{V}_{l,a}$$

where the previous layer's output $\mathbf{H}^{l-1} \in \mathbb{R}^{|x| \times d_h}$ is linearly projected to a triple of queries, keys and values using parameter matrices $\mathbf{W}_{l,a}^Q, \mathbf{W}_{l,a}^K, \mathbf{W}_{l,a}^V \in \mathbb{R}^{d_h \times d_k}$, respectively. $\mathbf{A}_{l,a} \in \mathbb{R}^{|x| \times |x|}$ indicates the attention distributions, which is computed by the scaled dot-product of queries and keys. $A_h$ represents the number of self-attention heads. $d_k \times A_h$ is equal to $d_h$ in BERT.

## 2.2    Transformer Distillation

Knowledge distillation [17, 32] is to train the small student model $S$ on a transfer feature set with soft labels and intermediate representations provided by the large teacher model $T$. Knowledge distillation is modeled as minimizing the differences between teacher and student features:

$$\mathcal{L}_{\text{KD}} = \sum_{e \in \mathcal{D}} L(f^S(e), f^T(e))$$

Where $\mathcal{D}$ denotes the training data, $f^S(\cdot)$ and $f^T(\cdot)$ indicate the features of student and teacher models respectively, $L(\cdot)$ represents the loss function. The mean squared error (MSE) and KL-divergence are often used as loss functions.

For task-agnostic Transformer based LM distillation, soft target probabilities for masked language modeling predictions, embedding layer outputs, self-attention distributions and outputs (hidden states) of each Transformer layer of the teacher model are used as features to help the training of the student. For the intermediate representations of each Transformer layer, previous works [20, 38] often map each student layer to its corresponding teacher layer and perform layer-to-layer distillation.

## 3    Deep Self-Attention Distillation

Figure 1 gives an overview of the deep self-attention distillation. The key idea is three-fold. First, we propose to train the student by deeply mimicking the self-attention module, which is the vital

Table 1: Comparison with previous task-agnostic Transformer based LM distillation approaches.

| Approach | Teacher Model | Distilled Knowledge | Layer-to-Layer Distillation | Requirements on the number of layers of students | Requirements on the hidden size of students |
|---|---|---|---|---|---|
| DistilBERT | BERT$_{\text{BASE}}$ | Soft target probabilities<br>Embedding outputs | | | ✓ |
| TinyBERT | BERT$_{\text{BASE}}$ | Embedding outputs<br>Hidden states<br>Self-Attention distributions | ✓ | | |
| MOBILEBERT | IB-BERT$_{\text{LARGE}}$ | Soft target probabilities<br>Hidden states<br>Self-Attention distributions | ✓ | ✓ | ✓ |
| MINILM | BERT$_{\text{BASE}}$ | Self-Attention distributions<br>Self-Attention value relation | | | |

component in the Transformer, of the teacher's last layer. Second, we introduce transferring the relation between values (i.e., the scaled dot-product between values) to achieve a deeper mimicry, in addition to performing attention distributions (i.e., the scaled dot-product of queries and keys) transfer in the self-attention module. Moreover, we show that introducing a teacher assistant [26] also helps the distillation of large pre-trained Transformer models when the size gap between the teacher model and student model is large.

## 3.1 Self-Attention Distribution Transfer

The attention mechanism [3] has been a highly successful neural network component for NLP tasks, which is also crucial for pre-trained LMs. Some works show that self-attention distributions of pre-trained LMs capture a rich hierarchy of linguistic information [19, 8]. Transferring self-attention distributions has been used in previous works for Transformer distillation [20, 38, 1]. We also utilize the self-attention distributions to help the training of the student. Specifically, we minimize the KL-divergence between the self-attention distributions of the teacher and student:

$$\mathcal{L}_{\text{AT}} = \frac{1}{A_h|x|}\sum_{a=1}^{A_h}\sum_{t=1}^{|x|} D_{KL}(\mathbf{A}_{L,a,t}^T \parallel \mathbf{A}_{M,a,t}^S)$$

Where $|x|$ and $A_h$ represent the sequence length and the number of attention heads. $L$ and $M$ represent the number of layers for the teacher and student. $\mathbf{A}_L^T$ and $\mathbf{A}_M^S$ are the attention distributions of the last Transformer layer for the teacher and student, respectively. They are computed by the scaled dot-product of queries and keys.

Different from previous works which transfer teacher's knowledge layer-to-layer, we only use the attention maps of the teacher's last layer. Jawahar et al. [19] show that top layers of BERT encode semantic features and capture long-distance dependency knowledge. These knowledge is more important for most downstream tasks. Besides, layer-to-layer transfer sets a tight restriction for each student layer. Ablation studies show that relaxing restrictions of layer mapping on student models improves performance. Distilling knowledge of the last Transformer layer also allows more flexibility for the number of layers of our student models, avoids the effort of finding the best layer mapping.

## 3.2 Self-Attention Value-Relation Transfer

In self-attention module, queries, keys, and values are the most basic and important vectors. The knowledge of queries and keys is transferred via attention distributions. To achieve a deeper mimicry of the self-attention module, we introduce the values and transfer the value relation. The value relation is computed via the multi-head scaled dot-product between values. The KL-divergence between the value relation of the teacher and student is used as the training objective:

$$\mathbf{VR}_{L,a}^T = \text{softmax}(\frac{\mathbf{V}_{L,a}^T \mathbf{V}_{L,a}^{T\mathsf{T}}}{\sqrt{d_k}}), \ \mathbf{VR}_{M,a}^S = \text{softmax}(\frac{\mathbf{V}_{M,a}^S \mathbf{V}_{M,a}^{S\mathsf{T}}}{\sqrt{d_k'}})$$

$$\mathcal{L}_{\text{VR}} = \frac{1}{A_h|x|}\sum_{a=1}^{A_h}\sum_{t=1}^{|x|} D_{KL}(\mathbf{VR}_{L,a,t}^T \parallel \mathbf{VR}_{M,a,t}^S)$$

Table 2: Comparison between the publicly released 6-layer models with 768 hidden size distilled from BERT$_{\text{BASE}}$. We compare task-agnostic distilled models without task-specific distillation and data augmentation. We report F1 for SQuAD 2.0, Matthews correlation coefficient for CoLA, and accuracy for other datasets. The GLUE results of DistilBERT are taken from Sanh et al. [35]. We report the SQuAD 2.0 result by fine-tuning their released model[5]. For TinyBERT, we fine-tune the latest version of their public model[6] for a fair comparison. We also report the fine-tuning results of Truncated BERT$_{\text{BASE}}$ and the 6x768 BERT model (BERT$_{\text{SMALL}}$) [41] trained using the MLM objective as baselines. Sajjad et al. [33] show that top-layer dropping consistently outperforms other strategies when dropping 6 layers, so we drop top 6 layers from BERT$_{\text{BASE}}$ for truncated BERT$_{\text{BASE}}$. The fine-tuning results are an average of 4 runs.

| Model | #Param | SQuAD2 | MNLI-m | SST-2 | QNLI | CoLA | RTE | MRPC | QQP | Average |
|---|---|---|---|---|---|---|---|---|---|---|
| BERT$_{\text{BASE}}$ [12] (teacher) | 109M | 76.8 | 84.5 | 93.2 | 91.7 | 58.9 | 68.6 | 87.3 | 91.3 | 81.5 |
| BERT$_{\text{SMALL}}$ [41] | 66M | 73.2 | 81.8 | 91.2 | 89.8 | 53.5 | 67.9 | 84.9 | 90.6 | 79.1 |
| Truncated BERT$_{\text{BASE}}$ [12] | 66M | 69.9 | 81.2 | 90.8 | 87.9 | 41.4 | 65.5 | 82.7 | 90.4 | 76.2 |
| DistilBERT [35] | 66M | 70.7 | 82.2 | 91.3 | 89.2 | **51.3** | 59.9 | 87.5 | 88.5 | 77.6 |
| TinyBERT [20] | 66M | 73.1 | 83.5 | 91.6 | 90.5 | 42.8 | **72.2** | **88.4** | 90.6 | 79.1 |
| MINILM | 66M | **76.4** | **84.0** | **92.0** | **91.0** | 49.2 | 71.5 | **88.4** | **91.0** | **80.4** |

Where $\mathbf{V}_{L,a}^T \in \mathbb{R}^{|x| \times d_k}$ and $\mathbf{V}_{M,a}^S \in \mathbb{R}^{|x| \times d'_k}$ are the values of an attention head in self-attention module for the teacher's and student's last layer. $\mathbf{VR}_L^T \in \mathbb{R}^{A_h \times |x| \times |x|}$ and $\mathbf{VR}_M^S \in \mathbb{R}^{A_h \times |x| \times |x|}$ are the value relation of the last Transformer layer for teacher and student, respectively. The final training loss is computed via summing the attention distribution and value-relation transfer losses.

Using value relation enables the student to deeply mimic the teacher's self-attention behavior. Compared with directly transferring value vectors, using scaled dot-product converts teacher and student value vectors of different dimensions into relation matrices with the same size. It avoids introducing additional parameters (randomly initialized) to linearly transform student's vectors into the same size as its teacher. The additional transformation transforms student vectors into another vector space and restricts teacher from directly transferring knowledge. Value relation also introduces more knowledge of word dependencies.

### 3.3 Teacher Assistant

Following Mirzadeh et al. [26], we introduce a teacher assistant (i.e., intermediate-size student model) to further improve the model performance of smaller students.

Assuming the teacher model consists of $L$-layer Transformer with $d_h$ hidden size, the student model has $M$-layer Transformer with $d'_h$ hidden size. For smaller students ($M \leq \frac{1}{2}L$, $d'_h \leq \frac{1}{2}d_h$), we first distill the teacher into a teacher assistant with $L$-layer Transformer and $d'_h$ hidden size. The assistant model is then used as the teacher to guide the training of the final student. The introduction of a teacher assistant bridges the size gap between teacher and smaller student models, helps the distillation of Transformer based pre-trained LMs. Moreover, combining deep self-attention distillation with a teacher assistant brings further improvements for smaller student models.

### 3.4 Comparison with Previous Work

Table 1 presents the comparison with previous approaches [35, 20, 38]. MOBILEBERT proposes using a specially designed inverted bottleneck model, which has the same model size as BERT$_{\text{LARGE}}$, as the teacher. The other methods utilize BERT$_{\text{BASE}}$ to conduct experiments. For the knowledge used for distillation, our method introduces the scaled dot-product between values in the self-attention module as the new knowledge to deeply mimic teacher's self-attention behavior. TinyBERT and MOBILEBERT transfer knowledge of the teacher to the student layer-to-layer. MOBILEBERT assumes the student has the same number of layers as its teacher. TinyBERT employs a uniform strategy to determine its layer mapping. DistilBERT initializes the student with teacher's parameters, therefore selecting layers of the teacher model is still needed. MINILM distills the self-attention knowledge of the teacher's last Transformer layer, which allows the flexible number of layers for the students and alleviates the effort of finding the best layer mapping. Student hidden size of DistilBERT and MOBILEBERT is required to be the same as its teacher. Bottleneck and inverted bottleneck

Table 3: Comparison between student models of different architectures distilled from BERT$_{\text{BASE}}$. $M$ and $d'_h$ indicate the number of layers and hidden dimension of the student model. TA indicates teacher assistant[6]. The fine-tuning results are averaged over 4 runs.

| Architecture | #Param | Model | SQuAD 2.0 | MNLI-m | SST-2 | Average |
|---|---|---|---|---|---|---|
| $M$=6;$d'_h$=384 | 22M | MLM-KD (Soft-Label Distillation) | 67.9 | 79.6 | 89.8 | 79.1 |
| | | TinyBERT | 71.6 | 81.4 | 90.2 | 81.1 |
| | | MINILM | 72.4 | 82.2 | 91.0 | 81.9 |
| | | MINILM (w/ TA) | **72.7** | **82.4** | **91.2** | **82.1** |
| $M$=4;$d'_h$=384 | 19M | MLM-KD (Soft-Label Distillation) | 65.3 | 77.7 | 88.8 | 77.3 |
| | | TinyBERT | 66.7 | 79.2 | 88.5 | 78.1 |
| | | MINILM | 69.4 | 80.3 | 90.2 | 80.0 |
| | | MINILM (w/ TA) | **69.7** | **80.6** | **90.6** | **80.3** |
| $M$=3;$d'_h$=384 | 17M | MLM-KD (Soft-Label Distillation) | 59.9 | 75.2 | 88.0 | 74.4 |
| | | TinyBERT | 63.6 | 77.4 | 88.4 | 76.5 |
| | | MINILM | 66.2 | 78.8 | 89.3 | 78.1 |
| | | MINILM (w/ TA) | **66.9** | **79.1** | **89.7** | **78.6** |

Table 4: The number of Embedding (Emd) and Transformer (Trm) parameters, and inference time for different models.

| #Layers | Hidden Size | #Param (Emd) | #Param (Trm) | Inference Time |
|---|---|---|---|---|
| 12 | 768 | 23.4M | 85.1M | 93.1s (1.0×) |
| 6 | 768 | 23.4M | 42.5M | 46.9s (2.0×) |
| 12 | 384 | 11.7M | 21.3M | 34.8s (2.7×) |
| 6 | 384 | 11.7M | 10.6M | 17.7s (5.3×) |
| 4 | 384 | 11.7M | 7.1M | 12.0s (7.8×) |
| 3 | 384 | 11.7M | 5.3M | 9.2s (10.1×) |

Table 5: Effectiveness of self-attention value-relation (Value-Rel) transfer. The fine-tuning results are averaged over 4 runs.

| Architecture | Model | SQuAD2 | MNLI-m | SST-2 |
|---|---|---|---|---|
| $M$=6;$d'_h$=384 | MINILM | **72.4** | **82.2** | **91.0** |
| | -Value-Rel | 71.0 | 80.9 | 89.9 |
| $M$=4;$d'_h$=384 | MINILM | **69.4** | **80.3** | **90.2** |
| | -Value-Rel | 67.5 | 79.0 | 89.2 |
| $M$=3;$d'_h$=384 | MINILM | **66.2** | **78.8** | **89.3** |
| | -Value-Rel | 64.2 | 77.8 | 88.3 |

modules are introduced in MOBILEBERT to keep the hidden size of the teacher and student are the same. TinyBERT uses an additional parameter matrix to transform student hidden states with smaller dimensions during distillation. Using value relation allows our students to use arbitrary hidden size without introducing additional transformation.

# 4 Experiments

We conduct monolingual and multilingual distillation experiments in different parameter size of student models.

## 4.1 Distillation Setup

We use the uncased version of BERT$_{\text{BASE}}$ as our teacher. BERT$_{\text{BASE}}$ [12] is a 12-layer Transformer with 768 hidden size, and 12 attention heads, which contains about 109M parameters. We use documents of English Wikipedia[3] and BookCorpus [49] for the pre-training data, following the preprocess and the WordPiece tokenization of Devlin et al. [12]. For the training of multilingual MINILM, we use XLM-R$_{\text{BASE}}$[4] [10] as the teacher, which is also a 12-layer Transformer with 768 hidden size. We perform knowledge distillation using the same training corpora as XLM-R$_{\text{BASE}}$.

## 4.2 English MINILM

### 4.2.1 Downstream Tasks

Following previous language model pre-training [12, 25] and task-agnostic pre-trained LM distillation [35, 20, 38], we evaluate on the extractive question answering and GLUE benchmark.

Table 6: Comparison between different loss functions: KL-divergence over the value relation and mean squared error (MSE) over value vectors. An additional parameter matrix is introduced to transform student value vectors to have the same dimensions as the teacher values. The fine-tuning results are an average of 4 runs for each task.

| Architecture | Model | SQuAD2 | MNLI-m | SST-2 |
|---|---|---|---|---|
| $M=6;d'_h=384$ | MINILM | **72.4** | **82.2** | **91.0** |
| | Value-MSE | 71.4 | 82.0 | 90.8 |
| $M=4;d'_h=384$ | MINILM | **69.4** | **80.3** | **90.2** |
| | Value-MSE | 68.3 | 80.1 | 89.9 |
| $M=3;d'_h=384$ | MINILM | **66.2** | **78.8** | **89.3** |
| | Value-MSE | 65.5 | 78.4 | **89.3** |

Table 7: Comparison between distilling knowledge of the teacher's last Transformer layer and layer-to-layer (La-to-La) distillation. We adopt a uniform strategy as in Jiao et al. [20] to determine the mapping between teacher and student layers. The fine-tuning results are an average of 4 runs for each task.

| Architecture | Model | SQuAD2 | MNLI-m | SST-2 |
|---|---|---|---|---|
| $M=6;d'_h=384$ | MINILM | **72.4** | **82.2** | **91.0** |
| | +La-to-La | 71.6 | 81.8 | 90.6 |
| $M=4;d'_h=384$ | MINILM | **69.4** | **80.3** | **90.2** |
| | +La-to-La | 67.6 | 79.9 | 89.6 |
| $M=3;d'_h=384$ | MINILM | **66.2** | **78.8** | **89.3** |
| | +La-to-La | 64.8 | 77.7 | 88.6 |

**Extractive Question Answering**   Given a passage $P$, the task is to select a contiguous span of text in the passage by predicting its start and end positions to answer the question $Q$. We evaluate on SQuAD 2.0 [31], which has served as a major question answering benchmark.

**GLUE**   The General Language Understanding Evaluation benchmark [44] consists of nine sentence-level classification tasks, including linguistic acceptability [45], sentiment analysis [36], text similarity [7], paraphrase detection [13], and natural language inference (NLI) [11, 5, 15, 6, 23, 46, 30].

### 4.2.2   Main Results

Previous works [35, 37, 20, 1] usually distill BERT_BASE into a 6-layer model with 768 hidden size. We first conduct distillation experiments using the same student architecture. Results on SQuAD 2.0 and GLUE dev sets are presented in Table 2. Since MOBILEBERT distills a specially designed teacher (in the BERT_LARGE size) with inverted bottleneck modules into a 24-layer student using the bottleneck modules, we do not compare with MOBILEBERT. MINILM outperforms DistilBERT[5], TinyBERT[6] and two BERT baselines across most tasks. Our model exceeds the two state-of-the-art distilled models by 3.0+% F1 on SQuAD 2.0. We present the inference time for models in different parameter size in Table 4. Our 6-layer 768-dimensional student model is $2.0\times$ faster than BERT_BASE, while retaining more than 99% performance on a variety of tasks, such as SQuAD 2.0 and MNLI.

We also conduct experiments for smaller student models. We compare MINILM with our implemented MLM-KD (knowledge distillation using soft target probabilities for masked language modeling predictions) and TinyBERT, which are trained using the same data and hyper-parameters. The results on SQuAD 2.0, MNLI and SST-2 dev sets are shown in Table 3. MINILM outperforms soft label distillation and our implemented TinyBERT on the three tasks. Deep self-attention distillation is also effective for smaller models. Moreover, we show that introducing a teacher assistant[7] is also helpful in Transformer based pre-trained LM distillation, especially for smaller models. Combining deep self-attention distillation with a teacher assistant achieves further improvement for smaller students.

We also conduct distillation experiments using an in-house pre-trained Transformer model following UNILM [14, 4] in the BERT_BASE size. The distilled models achieve better performance on natural language understanding tasks. The models can also be applied for natural language generation tasks and achieve competitive performance. Please refer to the supplementary material.

### 4.2.3   Ablation Studies

We do ablation tests on several tasks to analyze the contribution of self-attention value-relation transfer. The dev results of SQuAD 2.0, MNLI and SST-2 are illustrated in Table 5, using self-attention value-relation transfer positively contributes to the final results for student models in different parameter size. Distilling the fine-grained knowledge of value relation helps the student model deeply mimic the self-attention behavior of the teacher, which further improves model performance.

Table 8: Cross-lingual classification results of our 12-layer[a] and 6-layer[b] multilingual models with 384 hidden size on XNLI. We report the average accuracy. Results of mBERT, XLM-100 and XLM-R$_{\text{BASE}}$ are from Conneau et al. [10].

| Model | #Layers | #Hidden | Average |
|---|---|---|---|
| mBERT | 12 | 768 | 66.3 |
| XLM-100 | 16 | 1280 | 70.7 |
| XLM-R$_{\text{BASE}}$ | 12 | 768 | 74.5 |
| MiniLM[a] | 12 | 384 | 71.1 |
| MiniLM[b] (w/ TA) | 6 | 384 | 68.0 |

Table 9: Cross-lingual question answering results on MLQA. We report the average F1 and EM (exact match) scores. Results of mBERT and XLM-15 are taken from Lewis et al. [24]. † indicates results taken from Conneau et al. [10]. We also report our fine-tuned results (‡) of XLM-R$_{\text{BASE}}$.

| Model | #Layers | #Hidden | Average |
|---|---|---|---|
| mBERT | 12 | 768 | 57.7 / 41.6 |
| XLM-15 | 12 | 1024 | 61.6 / 43.5 |
| XLM-R$_{\text{BASE}}$† | 12 | 768 | 62.9 / 45.7 |
| XLM-R$_{\text{BASE}}$‡ | 12 | 768 | 64.9 / 46.9 |
| MiniLM[a] | 12 | 384 | 63.2 / 44.7 |
| MiniLM[b] (w/ TA) | 6 | 384 | 53.7 / 36.6 |

We compare our proposed value relation with directly transferring value vectors (using mean squared error (MSE)). An additional parameter matrix is introduced to transform student value vectors if the hidden dimension of the student is smaller than its teacher. The dev results on three tasks are presented in Table 6. Using value relation achieves better performance. Specifically, our method brings about $1.0\%$ F1 improvement on the SQuAD benchmark. Compared with transferring value vectors using MSE function, our method does not need to introduce additional transformations, which restrict teacher from directly transferring knowledge. We also show that transferring value relation performs better than transferring hidden states. Please refer to the supplementary material.

To show the effectiveness of distilling self-attention knowledge of the teacher's last Transformer layer, we compare our method with layer-to-layer distillation. We transfer the same knowledge and adopt a uniform strategy as in Jiao et al. [20] to perform layer-to-layer distillation. The dev results on three tasks are presented in Table 7. Using the last layer achieves better results. It also alleviates the difficulties in layer mapping between the teacher and student. Besides, distilling the teacher's last Transformer layer requires less computation than layer-to-layer distillation, results in faster training speed. To study why only using the last layer is more effective, we try more strategies to select layers and find that reducing the guidance of teacher's knowledge improves performance. Layer-to-layer transfer sets a tight restriction for each student layer. Transferring the last layer is more flexible and effective. Please refer to the supplementary material.

### 4.3 Multilingual MiniLM

We conduct experiments on multilingual pre-trained models and distill XLM-R$_{\text{BASE}}$ into 12-layer and 6-layer models with 384 hidden size. We evaluate the student models on cross-lingual natural language inference (XNLI) benchmark [9] and cross-lingual question answering (MLQA) benchmark [24].

**XNLI** Table 8 presents XNLI results of MiniLM and several pre-trained LMs. Following Conneau et al. [10], we select the best single model on the joint dev set of all the languages. MiniLM achieves competitive performance on XNLI with much fewer Transformer parameters. The 12x384 MiniLM compares favorably with mBERT [12] and XLM [22] trained on the MLM objective.

**MLQA** Following Lewis et al. [24], we adopt SQuAD 1.1 [30] as training data and use MLQA English development data for early stopping. As shown in Table 9, the 12x384 MiniLM performs competitively better than mBERT and XLM. 6-layer MiniLM also achieves competitive performance.

### 4.4 Discussion

MiniLM distills self-attention knowledge of the teacher's last Transformer layer. Previous works [19] show that BERT encodes surface features or phrase-level information at the bottom layers, syntactic features in the middle and semantic features at the top layers. Can the small model learn surface features or phrase-level information by only distilling the last Transformer layer? Following Jawahar et al. [19], we extract span representations from different layers to analyze the phrase-level information captured by our student model. We randomly pick labeled chunks and unlabeled spans from the CoNLL 2000 chunking dataset [34], and extract span representations from each layer of the 6x384

Table 10: Clustering results of span representations from different layers of BERT$_{\text{BASE}}$ and 6x384 MINILM (distilled from BERT$_{\text{BASE}}$). Results of BERT$_{\text{BASE}}$ are taken from Jawahar et al. [19]. Normalized mutual information is used to evaluate the clusters.

| Layer | 1 | 2 | 3 | 4 | 5 | 6 | 8 | 10 | 12 |
|-------|------|------|------|------|------|------|------|------|------|
| BERT$_{\text{BASE}}$ | 0.38 | 0.37 | 0.35 | 0.30 | 0.24 | 0.20 | 0.16 | 0.18 | 0.19 |
| MINILM | 0.45 | 0.49 | 0.46 | 0.47 | 0.37 | 0.36 | - | - | - |

model distilled from BERT$_{\text{BASE}}$. $k$-means clustering is then performed on span representations. As shown in Table 10, MINILM distilled from teacher's last Transformer layer also learns the phrase-level information well. Moreover, lower layers of MINILM also encode phrasal information better than higher layers.

We visualize the attention distributions[8] of the teacher (BERT$_{\text{BASE}}$) and the 6x384 students trained using last layer and layer-wise distillation (Figure 1-3 in supplementary material). We find that attention distributions for each layer of the layer-wise distilled student are very similar to its corresponding layers of the teacher. For the 6x384 student distilled from teacher's last layer, attention distributions of the last layer mimic its teacher's last layer very well, while the bottom layers are more similar to the teacher's bottom layers. Last layer distillation also learns features of teacher's lower layers.

## 5  Conclusion

In this work, we propose a simple and effective knowledge distillation method to compress large pre-trained Transformer based language models. The student is trained by deeply mimicking the teacher's self-attention modules, which are the vital components of the Transformer networks. We propose using the self-attention distributions and value relation of the teacher's last Transformer layer to guide the training of the student, which is effective and flexible for the student models. Moreover, we show that introducing a teacher assistant also helps pre-trained Transformer based LM distillation. Our student model distilled from BERT$_{\text{BASE}}$ retains high accuracy on SQuAD 2.0 and the GLUE benchmark tasks, and outperforms state-of-the-art baselines. Multilingual MINILM distilled from XLM-R$_{\text{BASE}}$ also achieves competitive performance. The deep self-attention distillation can also be applied to compress pre-trained models in larger size. We leave it as our future work.

## Broader Impact

Pre-trained language models have achieved remarkable success for various natural language processing tasks. However, these models consist of hundreds of millions of parameters and become bigger and bigger. It brings challenges for online serving in real-life applications due to latency and capacity constraints. Our work focuses on compressing large pre-trained models into small and fast pre-trained models, while achieving competitive performance. Our method and released models can be useful for a lot of real-life applications. Besides, fine-tuning large pre-trained models has hard requirements of GPU resources and the computational cost is also very high. Fine-tuning and running inference using small models can save GPU hours, dollars, and carbon dioxide emissions. Our small and fast models can also help researchers with less computing resources.

## Footnotes

[3]Wikipedia version: enwiki-20181101.

[4]We use the v0 version of XLM-R$_{\text{BASE}}$ in our distillation and fine-tuning experiments.

[5]`https://github.com/huggingface/transformers/tree/master/examples/distillation`

[6]`https://github.com/huawei-noah/Pretrained-Language-Model/tree/master/TinyBERT`

[7]The teacher assistant is only introduced for the model MINILM (w/ TA).

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
