[Supplementary Material]

# Supplementary Material: Deep Self-Attention Distillation for Task-Agnostic Compression of Pre-Trained Transformers

**Wenhui Wang     Furu Wei**[*]   **Li Dong    Hangbo Bao    Nan Yang    Ming Zhou**
Microsoft Research
{wenwan,fuwei,lidong1,t-habao,nanya,mingzhou}@microsoft.com

## 1   Supplementary Experiments

### 1.1   English MINILM Distilled from Better Teacher

We report the results of MINILM distilled from an in-house pre-trained Transformer model following UNILM [6, 3] in the BERT$_{BASE}$ size. The teacher model is trained using similar pre-training datasets as in RoBERTa$_{BASE}$ [20], which includes 160GB text corpora from English Wikipedia, BookCorpus [45], OpenWebText[2], CC-News [20], and Stories [36]. We distill the teacher model into 12-layer and 6-layer models with $384$ hidden size using the same corpora. The 12x384 model is used as the teacher assistant to train the 6x384 model.

#### 1.1.1   Results on NLU Tasks

We first evaluate MINILM on natural language understanding tasks. Table 1 presents the dev results of SQuAD 2.0 and GLUE benchmark. The results of 6x384 MINILM are significantly improved. The 12x384 MINILM achieves $2.7\times$ speedup while performing competitively better than BERT$_{BASE}$ on SQuAD 2.0 and GLUE benchmark.

#### 1.1.2   Results on NLG Tasks

We also show that MINILM can be applied for natural language generation tasks, such as question generation and abstractive summarization. Following Dong et al. [6], we fine-tune MINILM as a sequence-to-sequence model by employing a specific self-attention mask.

**Question Generation**   We conduct experiments for the answer-aware question generation task [7]. Given an input passage and an answer, the task is to generate a question that asks for the answer. The SQuAD 1.1 dataset [26] is used for evaluation. The results of MINILM, UNILM$_{LARGE}$ and several state-of-the-art models are presented in Table 2, our 12x384 and 6x384 distilled models achieve competitive performance on the question generation task.

**Abstractive Summarization**   We evaluate MINILM on two abstractive summarization datasets, i.e., XSum [22], and the non-anonymized version of CNN/DailyMail [30]. The generation task is to condense a document into a concise and fluent summary, while conveying its key information. We report ROUGE scores [18] on the datasets. Table 3 presents the results of MINILM, baseline, several state-of-the-art models and pre-trained Transformer models. Our 12x384 model outperforms BERT based method BERTSUMABS [19] and the pre-trained sequence-to-sequence model MASS$_{BASE}$ [31] with much fewer parameters. Moreover, our 6x384 MINILM also achieves competitive performance.

---

[*]  Contact person.
[2]skylion007.github.io/OpenWebTextCorpus

Table 1: The results of MINILM distilled from an in-house pre-trained Transformer model (BERT$_{BASE}$ size, 12-layer Transformer, 768-hidden size, and 12 self-attention heads) on SQuAD 2.0 and GLUE benchmark. We report the results of our 12-layer[a] and 6-layer[b] models with 384 hidden size. The fine-tuning results are averaged over 4 runs.

| Model | #Param | SQuAD2 | MNLI-m | SST-2 | QNLI | CoLA | RTE | MRPC | QQP | Average |
|---|---|---|---|---|---|---|---|---|---|---|
| BERT$_{BASE}$ | 109M | 76.8 | 84.5 | 93.2 | 91.7 | 58.9 | 68.6 | 87.3 | 91.3 | 81.5 |
| MINILM[a] | 33M | 81.7 | 85.7 | 93.0 | 91.5 | 58.5 | 73.3 | 89.5 | 91.3 | 83.1 |
| MINILM[b] (w/ TA) | 22M | 75.6 | 83.3 | 91.5 | 90.5 | 47.5 | 68.8 | 88.9 | 90.6 | 79.6 |

Table 2: Question generation results of our 12-layer[a] and 6-layer[b] models with 384 hidden size on SQuAD 1.1. The first block follows the data split in Du and Cardie [7], while the second block is the same as in Zhao et al. [44].

| | #Param | BLEU-4 | METEOR | ROUGE-L |
|---|---|---|---|---|
| Du and Cardie [7] | | 15.16 | 19.12 | - |
| Zhang and Bansal [43] | | 18.37 | 22.65 | 46.68 |
| UNILM$_{LARGE}$ | 340M | 22.78 | 25.49 | 51.57 |
| MINILM[a] | 33M | 21.07 | 24.09 | 49.14 |
| MINILM[b] (w/ TA) | 22M | 20.31 | 23.43 | 48.21 |
| Zhao et al. [44] | | 16.38 | 20.25 | 44.48 |
| Zhang and Bansal [43] | | 20.76 | 24.20 | 48.91 |
| UNILM$_{LARGE}$ | 340M | 24.32 | 26.10 | 52.69 |
| MINILM[a] | 33M | 23.27 | 25.15 | 50.60 |
| MINILM[b] (w/ TA) | 22M | 22.01 | 24.24 | 49.51 |

## 1.2 Multilingual MINILM

We present the number of Transformer and embedding parameters for different multilingual pre-trained models and our distilled models in Table 4. We also report the XNLI results for each language in Table 5, MLQA results for each language in Table 6.

## 1.3 Supplementary Ablation Studies

Table 7 presents the comparison between transferring value relation and transferring hidden states using MSE. We use self-attention distributions and hidden states of teacher's last Transformer layer to guide the training of the student model (Hidden-MSE). A parameter matrix is introduced to transform student hidden states to have the same size as the teacher hidden states. Using value relation performs better than transferring hidden states. Transferring value relation avoids additional transformation and introduces more knowledge of word dependencies. We have also tried to transfer the relation between hidden states instead of directly transferring vectors. But we find the performance of student models are unstable for different teacher models.

To study the rationale behind transferring teacher's last Transformer layer, we compare more strategies of mapping teacher and student layers. We conduct experiments using a 3-layer student model with 384 hidden size. Besides transferring teacher's knowledge to the last student layer and all three student layers (adopt a uniform strategy to map each teacher and student layers), we use the uniform strategy to determine the mapping of teacher and student layers but only transfer teacher's knowledge of corresponding layers to the last two layers, first and last two layers of the student model. Table 8 shows the results of different strategies. Transferring the last layer performs better than the strategies using two layers. Transferring two layers achieves better performance than transferring all three layers. Relaxing restrictions of layer mapping improves performance. Given the student always has fewer number of layers, the knowledge ideally learned at each student layer may be different from the knowledge of corresponding layers of the teacher model. Only transferring teacher's last layer gives the student more flexibility to learn the knowledge.

Table 3: Abstractive summarization results of our 12-layer[a] and 6-layer[b] models with 384 hidden size on CNN/DailyMail and XSum. The evaluation metric is the F1 version of ROUGE (RG) scores.

| Model | #Param | CNN/DailyMail | | | XSum | | |
|---|---|---|---|---|---|---|---|
| | | RG-1 | RG-2 | RG-L | RG-1 | RG-2 | RG-L |
| LEAD-3 | | 40.42 | 17.62 | 36.67 | 16.30 | 1.60 | 11.95 |
| PTRNET [30] | | 39.53 | 17.28 | 36.38 | 28.10 | 8.02 | 21.72 |
| Bottom-Up [8] | | 41.22 | 18.68 | 38.34 | - | - | - |
| UNILM$_{LARGE}$ [6] | 340M | 43.08 | 20.43 | 40.34 | - | - | - |
| BART$_{LARGE}$ [16] | 400M | 44.16 | 21.28 | 40.90 | 45.14 | 22.27 | 37.25 |
| T5$_{11B}$ [25] | 11B | 43.52 | 21.55 | 40.69 | - | - | - |
| MASS$_{BASE}$ [31] | 123M | 42.12 | 19.50 | 39.01 | 39.75 | 17.24 | 31.95 |
| BERTSUMABS [19] | 156M | 41.72 | 19.39 | 38.76 | 38.76 | 16.33 | 31.15 |
| T5$_{BASE}$ [25] | 220M | 42.05 | 20.34 | 39.40 | - | - | - |
| MINILM$^a$ | 33M | 42.66 | 19.91 | 39.73 | 40.43 | 17.72 | 32.60 |
| MINILM$^b$ (w/ TA) | 22M | 41.57 | 19.21 | 38.64 | 38.79 | 16.39 | 31.10 |

Table 4: The number of Transformer (Trm) and Embedding (Emd) parameters for different multilingual pre-trained models and our distilled models.

| Model | #Layers | Hidden Size | #Vocab | #Param (Trm) | #Param (Emd) | Speedup |
|---|---|---|---|---|---|---|
| mBERT | 12 | 768 | 110k | 85M | 85M | 1.0× |
| XLM-15 | 12 | 1024 | 95k | 151M | 97M | 0.6× |
| XLM-100 | 16 | 1280 | 200k | 315M | 256M | 0.3× |
| XLM-R$_{BASE}$ | 12 | 768 | 250k | 85M | 192M | 1.0× |
| MINILM$^a$ | 12 | 384 | 250k | 21M | 96M | 2.7× |
| MINILM$^b$ | 6 | 384 | 250k | 11M | 96M | 5.3× |

## 2 Related Work

### 2.1 Pre-trained Language Models

Unsupervised pre-training of language models [23, 10, 24, 5, 2, 31, 6, 41, 13, 20, 16, 25, 15, 4] has achieved significant improvements for a wide range of NLP tasks. Early methods for pre-training [23, 24] were based on standard language models. Recently, BERT [5] proposes to use a masked language modeling objective to train a deep bidirectional Transformer encoder, which learns interactions between left and right context. Liu et al. [20] show that very strong performance can be achieved by training the model longer over more data. Joshi et al. [13] extend BERT by masking contiguous random spans. Yang et al. [41] predict masked tokens auto-regressively in a permuted order.

To extend the applicability of pre-trained Transformers for NLG tasks. Dong et al. [6] extend BERT by utilizing specific self-attention masks to jointly optimize bidirectional, unidirectional and sequence-to-sequence masked language modeling objectives. Raffel et al. [25] employ an encoder-decoder Transformer and perform sequence-to-sequence pre-training by predicting the masked tokens in the encoder and decoder. Different from Raffel et al. [25], Lewis et al. [16] predict tokens auto-regressively in the decoder.

### 2.2 Knowledge Distillation

Knowledge distillation has proven a promising way to compress large models while maintaining accuracy. It transfers the knowledge of a large model or an ensemble of neural networks (teacher) to a single lightweight model (student). Hinton et al. [9] first propose transferring the knowledge of the teacher to the student by using its soft target distributions to train the distilled model. Romero et al. [28] introduce intermediate representations from hidden layers of the teacher to guide the training of the student. Knowledge of the attention maps [42, 11] is also introduced to help the training.

In this work, we focus on task-agnostic knowledge distillation of large pre-trained Transformer based language models. There have been some works that task-specifically distill the fine-tuned language models on downstream tasks. Tang et al. [35] distill fine-tuned BERT into an extremely small bidirectional LSTM. Turc et al. [38] initialize the student with a small pre-trained LM during

Table 5: Cross-lingual classification results of our 12-layer[a] and 6-layer[b] multilingual models with 384 hidden size on XNLI. We report the accuracy on each of the 15 XNLI languages and the average accuracy. Results of mBERT, XLM-100 and XLM-R$_{\text{BASE}}$ are from Conneau et al. [4].

| Model | #Layers | #Hidden | en | fr | es | de | el | bg | ru | tr | ar | vi | th | zh | hi | sw | ur | Avg |
|---|---|---|---|---|---|---|---|---|---|---|---|---|---|---|---|---|---|---|
| mBERT | 12 | 768 | 82.1 | 73.8 | 74.3 | 71.1 | 66.4 | 68.9 | 69.0 | 61.6 | 64.9 | 69.5 | 55.8 | 69.3 | 60.0 | 50.4 | 58.0 | 66.3 |
| XLM-100 | 16 | 1280 | 83.2 | 76.7 | 77.7 | 74.0 | 72.7 | 74.1 | 72.7 | 68.7 | 68.6 | 72.9 | 68.9 | 72.5 | 65.6 | 58.2 | 62.4 | 70.7 |
| XLM-R$_{\text{BASE}}$ | 12 | 768 | 84.6 | 78.4 | 78.9 | 76.8 | 75.9 | 77.3 | 75.4 | 73.2 | 71.5 | 75.4 | 72.5 | 74.9 | 71.1 | 65.2 | 66.5 | 74.5 |
| MINILM[a] | 12 | 384 | 81.5 | 74.8 | 75.7 | 72.9 | 73.0 | 74.5 | 71.3 | 69.7 | 68.8 | 72.1 | 67.8 | 70.0 | 66.2 | 63.3 | 64.2 | 71.1 |
| MINILM[b] (w/ TA) | 6 | 384 | 79.2 | 72.3 | 73.1 | 70.3 | 69.1 | 72.0 | 69.1 | 64.5 | 64.9 | 69.0 | 66.0 | 67.8 | 62.9 | 59.0 | 60.6 | 68.0 |

Table 6: Cross-lingual question answering results of our 12-layer[a] and 6-layer[b] multilingual models with 384 hidden size on MLQA. We report the F1 and EM (exact match) scores on each of the 7 MLQA languages. Results of mBERT and XLM-15 are taken from Lewis et al. [17]. † indicates results of XLM-R$_{\text{BASE}}$ taken from Conneau et al. [4]. We also report our fine-tuned results (‡) of XLM-R$_{\text{BASE}}$.

| Model | #Layers | #Hidden | en | es | de | ar | hi | vi | zh | Avg |
|---|---|---|---|---|---|---|---|---|---|---|
| mBERT | 12 | 768 | 77.7 / 65.2 | 64.3 / 46.6 | 57.9 / 44.3 | 45.7 / 29.8 | 43.8 / 29.7 | 57.1 / 38.6 | 57.5 / 37.3 | 57.7 / 41.6 |
| XLM-15 | 12 | 1024 | 74.9 / 62.4 | 68.0 / 49.8 | 62.2 / 47.6 | 54.8 / 36.3 | 48.8 / 27.3 | 61.4 / 41.8 | 61.1 / 39.6 | 61.6 / 43.5 |
| XLM-R$_{\text{BASE}}$† | 12 | 768 | 77.8 / 65.3 | 67.2 / 49.7 | 60.8 / 47.1 | 53.0 / 34.7 | 57.9 / 41.7 | 63.1 / 43.1 | 60.2 / 38.0 | 62.9 / 45.7 |
| XLM-R$_{\text{BASE}}$‡ | 12 | 768 | 80.3 / 67.4 | 67.0 / 49.2 | 62.7 / 48.3 | 55.0 / 35.6 | 60.4 / 43.7 | 66.5 / 45.9 | 62.3 / 38.3 | 64.9 / 46.9 |
| MINILM[a] | 12 | 384 | 79.4 / 66.5 | 66.1 / 47.5 | 61.2 / 46.5 | 54.9 / 34.9 | 58.5 / 41.3 | 63.1 / 42.1 | 59.0 / 33.8 | 63.2 / 44.7 |
| MINILM[b] (w/ TA) | 6 | 384 | 75.5 / 61.9 | 55.6 / 38.2 | 53.3 / 37.7 | 43.5 / 26.2 | 46.9 / 31.5 | 52.0 / 33.1 | 48.8 / 27.3 | 53.7 / 36.6 |

task-specific distillation. Sun et al. [32] introduce the hidden states from every $k$ layers of the teacher to perform knowledge distillation layer-to-layer. Aguilar et al. [1] further introduce the knowledge of self-attention distributions and propose progressive and stacked distillation methods. Mukherjee and Awadallah [21] propose a stage-wise scheme to transfer teacher's intermediate representations. Task-specific distillation requires to first fine-tune the large pre-trained LMs on downstream tasks and then perform knowledge transfer. The procedure of fine-tuning large pre-trained LMs is costly and time-consuming, especially for large datasets.

For task-agnostic distillation, the distilled model mimics the original large pre-trained LM and can be directly fine-tuned on downstream tasks. In practice, task-agnostic compression of pre-trained LMs is more desirable. MiniBERT [37] uses the soft target distributions for masked language modeling predictions to guide the training of the multilingual student model and shows its effectiveness on sequence labeling tasks. DistilBERT [29] uses the soft label and embedding outputs of the teacher to train the student. TinyBERT [12] and MOBILEBERT [33] further introduce self-attention distributions and hidden states to train the student. MOBILEBERT employs inverted bottleneck and bottleneck modules for teacher and student to make their hidden dimensions the same. The student model of MOBILEBERT is required to have the same number of layers as its teacher to perform layer-to-layer distillation. Besides, MOBILEBERT proposes a bottom-to-top progressive scheme to transfer teacher's knowledge. TinyBERT uses a uniform-strategy to map the layers of teacher and student when they have different number of layers, and a linear matrix is introduced to transform the student hidden states to have the same dimensions as the teacher. TinyBERT also introduces task-specific distillation and data augmentation for downstream tasks, which brings further improvements.

Different from previous works, our method employs the self-attention distributions and value relation of the teacher's last Transformer layer to help the student deeply mimic the self-attention behavior of the teacher. Using knowledge of the last Transformer layer instead of layer-to-layer distillation avoids restrictions on the number of student layers and the effort of finding the best layer mapping. Distilling relation between self-attention values allows the hidden size of students to be more flexible and avoids introducing linear matrices to transform student representations.

## 3 Hyper-parameters for Distillation

For the distillation experiments using BERT$_{\text{BASE}}$ as the teacher, the number of heads of attention distributions and value relation are set to 12 for student models. The vocabulary size is $30,522$. The maximum sequence length is 512. We use Adam [14] with $\beta_1 = 0.9$, $\beta_2 = 0.999$. We train the 6-layer student model with 768 hidden size using 1024 as the batch size and 5e-4 as the peak learning

Table 7: Comparison between transferring value relation and transferring hidden states using MSE. The fine-tuning results are an average of $4$ runs for each task.

| Architecture | Model | SQuAD 2.0 | MNLI-m | SST-2 | Average |
|---|---|---|---|---|---|
| $M$=3;$d'_h$=384 | MINILM | **66.2** | **78.8** | **89.3** | **78.1** |
| | Hidden-MSE | 64.5 | 78.0 | 88.9 | 77.1 |

Table 8: Comparison between different strategies to select layers and transfer teacher's knowledge. We adopt a uniform strategy to determine the mapping of teacher and student layers, and transfer teacher's knowledge of corresponding layers to the last layer, last two layers, first and last two layers, and all three layers of the student model. The fine-tuning results are an average of $4$ runs for each task.

| Architecture | Model | SQuAD 2.0 | MNLI-m | SST-2 | Average |
|---|---|---|---|---|---|
| $M$=3;$d'_h$=384 | Last Layer | **66.2** | **78.8** | **89.3** | **78.1** |
| | Last Two Layers | 65.4 | 78.1 | 89.1 | 77.5 |
| | First and Last Layers | 65.4 | 78.4 | 88.5 | 77.4 |
| | All Three Layers | 64.8 | 77.7 | 88.6 | 77.0 |

rate for $400,000$ steps. For student models of other architectures, the batch size and peak learning rate are set to 256 and 3e-4, respectively. We use linear warmup over the first $4,000$ steps and linear decay. The dropout rate is $0.1$. The weight decay is $0.01$. All the student models are initialized randomly.

We also use an in-house pre-trained Transformer model in the $BERT_{BASE}$ size as the teacher model, and distill it into 12-layer and 6-layer student models with 384 hidden size. For the 12-layer model, we use Adam [14] with $\beta_1 = 0.9$, $\beta_2 = 0.98$. The model is trained using 2048 as the batch size and 6e-4 as the peak learning rate for $400,000$ steps. The batch size and peak learning rate are set to 512 and 4e-4 for the 6-layer model. The rest hyper-parameters are the same as above BERT based distilled models.

For the training of multilingual MINILM models, we use Adam [14] with $\beta_1 = 0.9$, $\beta_2 = 0.999$. We train the 12-layer student model using 256 as the batch size and 3e-4 as the peak learning rate for $1,000,000$ steps. The vocabulary size is $250,002$. The maximum sequence length is $512$. The 6-layer student model is trained using $512$ as the batch size and 6e-4 as the peak learning rate for $400,000$ steps.

We distill our student models using 8 V100 GPUs with mixed precision training. Following Sun et al. [32] and Jiao et al. [12], the inference time is evaluated on the QNLI training set with the same hyper-parameters. We report the average running time of 100 batches on a single P100 GPU.

## 4   Hyper-parameters for Fine-tuning

**Extractive Question Answering**   For SQuAD 2.0, the maximum sequence length is $384$ and a sliding window of size $128$ if the lengths are longer than $384$. For the 12-layer model distilled from our in-house pre-trained model, we fine-tune 3 epochs using $48$ as the batch size and 4e-5 as the peak learning rate. The rest distilled models are trained using $32$ as the batch size and 6e-5 as the peak learning rate for 3 epochs.

**GLUE**   The maximum sequence length is $128$ for the GLUE benchmark. We set batch size to 32, choose learning rates from {2e-5, 3e-5, 4e-5, 5e-5} and epochs from {3, 4, 5} for student models distilled from $BERT_{BASE}$. For student models distilled from our in-house pre-trained model, the batch size is chosen from {32, 48}. We fine-tune several tasks (CoLA, RTE and MRPC) with longer epochs (up to 10 epochs), which brings slight improvements. For the 12-layer model, the learning rate used for CoLA, RTE and MRPC tasks is 1.5e-5.

**Question Generation**   For the question generation task, we set batch size to 32, and total length to $512$. The maximum output length is $48$. The learning rates are 3e-5 and 8e-5 for the 12-layer and 6-layer models, respectively. They are both fine-tuned for 25 epochs. We also use label smoothing [34]

Table 9: Summary of the GLUE benchmark.

| Corpus | #Train | #Dev | #Test | Metrics |
|--------|--------|------|-------|---------|
| *Single-Sentence Tasks* | | | | |
| CoLA | 8.5k | 1k | 1k | Matthews Corr |
| SST-2 | 67k | 872 | 1.8k | Accuracy |
| *Similarity and Paraphrase Tasks* | | | | |
| QQP | 364k | 40k | 391k | Accuracy/F1 |
| MRPC | 3.7k | 408 | 1.7k | Accuracy/F1 |
| STS-B | 7k | 1.5k | 1.4k | Pearson/Spearman Corr |
| *Inference Tasks* | | | | |
| MNLI | 393k | 20k | 20k | Accuracy |
| RTE | 2.5k | 276 | 3k | Accuracy |
| QNLI | 105k | 5.5k | 5.5k | Accuracy |
| WNLI | 634 | 71 | 146 | Accuracy |

Table 10: Dataset statistics and metrics of SQuAD 2.0.

| #Train | #Dev | #Test | Metrics |
|--------|------|-------|---------|
| 130,319 | 11,873 | 8,862 | Exact Match/F1 |

with rate of $0.1$. During decoding, we use beam search with beam size of $5$. The length penalty [40] is $1.3$.

**Abstractive Summarization** For the abstractive summarization task, we set batch size to $64$, and the rate of label smoothing to $0.1$. For the CNN/DailyMail dataset, the total length is $768$ and the maximum output length is $160$. The learning rates are 1e-4 and 1.5e-4 for the 12-layer and 6-layer models, respectively. They are both fine-tuned for $25$ epochs. During decoding, we set beam size to $5$, and the length penalty to $0.7$. For the XSum dataset, the total length is $512$ and the maximum output length is $48$. The learning rates are 1e-4 and 1.5e-4 for the 12-layer and 6-layer models, respectively. We fine-tune $30$ epochs for the 12-layer model and $50$ epochs for the 6-layer model. During decoding, we use beam search with beam size of $5$. The length penalty is set to $0.9$.

**Cross-lingual Natural Language Inference** The maximum sequence length is $128$ for XNLI. We fine-tune $5$ epochs using $128$ as the batch size, choose learning rates from {3e-5, 4e-5, 5e-5, 6e-5}.

**Cross-lingual Question Answering** For MLQA, the maximum sequence length is $512$ and a sliding window of size $128$ if the lengths are longer than $512$. We fine-tune $3$ epochs using $32$ as the batch size. The learning rates are chosen from {3e-5, 4e-5, 5e-5, 6e-5}.

## 5 GLUE Benchmark

The summary of datasets used for the General Language Understanding Evaluation (GLUE) benchmark[3] [39] is presented in Table 9. We add a linear classifier on top of the [CLS] token to predict label probabilities.

## 6 SQuAD 2.0

We present the dataset statistics and metrics of SQuAD 2.0[4] [27] in Table 10. Following BERT [5], we pack the question and passage tokens together with special tokens, to form the input: "[CLS] $Q$ [SEP] $P$ [SEP]". Two linear output layers are introduced to predict the probability of each token being the start and end positions of the answer span. The questions that do not have an answer are treated as having an answer span with start and end at the [CLS] token.

Figure 1: The visualization of attention distributions of 6x384 student distilled from teacher's last layer.

## 7 Visualization of Attention Distributions

Figure 1–3 present the visualization of attention distributions of the teacher model (BERT$_{\text{BASE}}$) and the 6x384 student models. The input sequence is "Ronaldo became Manchester United's first-ever Portuguese player when he signed before the 2003–04 season.".

## Footnotes

[3]https://gluebenchmark.com/

[4]http://stanford-qa.com

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

Figure 3: The visualization of attention distributions of BERT$_{\text{BASE}}$.