[Reviews · NeurIPS 2020]

Review 1

Summary and Contributions: This paper addresses the problem of reducing the sizes of transformer based models by knowledge distillation from a large teacher model to a smaller student model. In contrast to earlier work that performs layer wise distillation, this paper only distills the final layer of the transformer. In order to overcome the size restrictions imposed by the hidden dimensions of the model, they propose to have a self dot product of the "values" in the self attention module which allows them to distill from and to models using any hidden dimension. They also show that using an intermediate step of distilling to a model with the same number of layers and smaller hidden dimension before the final step to the model with fewer layers (which they refer to as using a teacher assistant), helps performance. The proposed method is quite simple and straight forward.

Strengths: 1) Since this paper only distills the last layer of the teacher transformer model, the number of layers in the student model is not constrained (unlike in most other approaches that do layer-wise distillation). 2) They propose to use the values of the self attention module as additional information that is distilled. They get around the need to have projection matrices that make the hidden dimensions of the teacher and student the same by instead forming a value relation matrix that is hidden size agnostic. 3) The results look interesting and given that only the last layer is distilled and not even the hidden outputs, it is interesting to see the results are comparable to other methods that require more moving parts.

Weaknesses: The results seem not consistent with prior reported numbers so its a little hard to evaluate the contribution. Further, the paper mentions MobileBERT which has proven to give very good results with extreme compression but does not report the numbers in the table. Although the teacher model is different, I believe if the student model sizes are comparable, it is important to report the results of the similarly sized models available. In this case, MobileBERT with 25.3 M parameters (half of the MiniLM proposed in this approach) and MobileBERT-tiny has 15.1M parameters and the results from this should definitely be reported. The paper has dev set results for MNLI-m, QNLI, MRPC, SST-2 and SQuAD . It would have also been useful to see the result of using the last layer of the BERT-large model since one of the main contributions of this paper is that there is no constraint on the size of the teacher and student models, and they also have a way to deal with the possible difficulties by using the teacher assistant.

Correctness: The TinyBert paper states a dev score on COLA of 54, for the (Num layers=6;hidden_dim=768; ffn=3072) which is widely different from the 42.8 reported in Table 2. The smaller model having 14.5 M parameters also gets 49.7 according to TinyBert. For MNLI-m, the reported results in TinyBERT for the same number of parameters is 84.5 as compared to 83.5 reported in this paper. For QQP, TinyBERT reports 91.1 acc compared to 90.6 reported here. SST2 ;93 -> 91.6. I would like to request the authors to verify their numbers. Update: They responded with results.

Clarity: Clearly written paper.

Relation to Prior Work: Differences from previous work are clearly explained.

Reproducibility: Yes

Additional Feedback: Minor comments: Line 126 These knowledge -> This knowledge


Review 2

Summary and Contributions: Edit: I've read the author response. I hope that the authors will add the additional analysis as they mention, which would strengthen the paper. With this in mind, I have updated my score. This paper proposes to distill a large Transformer model into a smaller one by minimizing the difference between the attention and values in the self-attention module in the last layer of the two modules. In addition, they propose to distill from a distilled teacher to bridge the size difference as in a prior work. The proposed model is evaluated on GLUE where it outperforms the comparison methods.

Strengths: 1. The proposed method seems to work well while being conceptually simple. 2. The method is evaluated with different models as foundation (BERT-Base and XLM-R). 3. The authors perform an extensive set of ablation analyses. In particular, the additional results and ablations provided in the supplementary material are comprehensive.

Weaknesses: 1. The novelty of the proposed method seems somewhat limited. In contrast to previous work, it only distills the top layer of a Transformer. The only additional methodological contribution is to also regularize the values. 2. I would have liked to see an analysis to what extent representations of the lower-level layers differ between the distilled and the teacher model. This could be done with singular vector canonical correlation analysis (SVCCA) or probing on low-level tasks such as parsing or POS tagging. I think such an analysis would provide people with more intuition about the benefit of restricting the distillation to the last layer.

Correctness: 1. It is not clear to me how the authors arrived at the parameter sizes for the models shown in Table 2. According to the TinyBERT paper (https://arxiv.org/pdf/1909.10351.pdf; Table 3), TinyBERT consists of 14.5M parameters. The same number is also referenced in the MobileBERT paper (https://arxiv.org/pdf/2004.02984.pdf; Table 4). This is in contrast to the 66M indicated in Table 2. Could you clarify which TinyBERT model you used, i.e. whether it is the 6layer-768dim one? 2. I would have liked to see a comparison to MobileBERT. It would be good to see if the proposed method is competitive despite using a more general setup.

Clarity: The paper is well written and easy to follow.

Relation to Prior Work: The paper positions itself appropriately in relation to prior work.

Reproducibility: Yes

Additional Feedback: - How do you explain that only distilling the last layer works better than also distilling earlier layers given what we know about about the representations obtained through masked language modelling? In particular, the last layers are generally task-specific and often not that used for downstream tasks. For instance, this recent paper (https://arxiv.org/pdf/2006.05987.pdf) shows that re-initializing the last hidden layers of a pre-trained BERT model actually improves fine-tuning performance. On the whole, even though the proposed method is not entirely novel, I feel that its conceptual simplicity, the comprehensive ablation analyses, and the strong results add enough value to the community to warrant acceptance.


Review 3

Summary and Contributions: This paper proposes a new distillation method which the authors refer to a MiniLM. The key differentiating factor is the distillation of the self-attention module as opposed to only the logits. The authors show that this method is effective as it outperforms other distillation models at the same parameter cost. The idea is obvious but the paper is overall well executed. The self-attention distribution transfer is obvious. However, the value relation transfer is indeed surprising and probably the most novel part of this work. But it makes one wonder why not query-query or key-key relation but value-value relation? Is there a difference? Experiments are conducted on GLUE and Squad and the results look pretty good. The authors also conduct may ablation studies. I think the work is quite well executed although the idea is not terribly novel. Overall, this paper should be very useful for practitioners that are interested in using lightweight pretrained Transformers in production. I think this direction is an important one and it is indeed enlightening that distilling using the self-attention weights is helpful. I am voting for a weak accept for this paper. Although I don't actively do research in this area, I think the approach is quite important and pushes a useful direction for the community.

Strengths: The paper tackles an important problem and shows that attention transfer is useful in distillation. Extensive ablations are conducted and generally provides useful insights for practitioners and researchers.

Weaknesses: It would be useful to have some visual analysis of these attention weight transfer.

Correctness: Everything looks good to me.

Clarity: The paper is well written.

Relation to Prior Work: Yes.

Reproducibility: Yes

Additional Feedback:


Review 4

Summary and Contributions: The paper presents a task-agnostic distilled model of BERT. The authors used teacher-student model for distillation. Different from previous work on using KD, they distill the attention module of the last layer of the transformer model. They used dot-product between values in the attention module, in addition to the attention distribution. The proposed model is compared extensively using several standard tasks, and previously publised distilled models. The results are quite impressive with preserving 99% of the performance on SQUAD.

Strengths: The paper is nicely written and a comprehensive evaluation is done. Most of the results and experiments that I could think of, are present in the paper.

Weaknesses: No serious weakness

Correctness: Yes, claims are solid and well supported by the evidence.

Clarity: Yes

Relation to Prior Work: Most of the related work is covered. I found two additional papers that should be cited: Bert-of-theseus: Compressing BERT by progressive module replacing Poor Man’s BERT: Smaller and Faster Transformer Models

Reproducibility: Yes

Additional Feedback: It is remarkable and at the same time puzzling to see that by just using attention-values from the top layer, they are able to preserve most of the performance of the base model. I could not comprehend why by considering scaled dot product between self-attention values result in better performance than other models. I would have expected bottom layers to be more critical since they are closer to the input while higher layers capture very abstract concepts. A few other comments: - It is not clear how the student model is initialized? Researchers has used alternate layers of teacher, random initialization and lower layers of the base model. This has shown to impact the performance of the students. - [1] showed that simply using lower layers of the base model outperformed several KD-based distillation methods. Authors have also provided this as a baseline. Authors may cite [1] (poor man's BERT) to support their baseline as a strong baseline reported in the literature. - Authors mentioned on line 146, value relation also introduces more knowledge of word dependencies. It would be nice to support this with some analysis. [1] Poor Man’s BERT: Smaller and Faster Transformer Models

[Author Response · NeurIPS 2020]

# 1 Response to all Reviewers

Thank you for the valuable suggestions. We will cite missing relevant works and add the results as you suggest.

**Comparison to MobileBERT.**  We distill a base-sized teacher, which achieves similar performance as the specially designed teacher of MobileBERT, into a same-sized student (25.3M parameters) using the same training data as MobileBERT. We compare with MobileBERT w/o OPT (without operational optimizations) and compute the speedup over $BERT_{BASE}$ according to their reported latency. As shown in Table 1, MiniLM achieves competitive results with faster speed. Moreover, our method can be applied for different teachers and has much fewer restrictions of students.

| Model | #Params | Speedup over $BERT_{BASE}$ | SQuAD 2.0 | MNLI-m | QNLI | SST-2 | MRPC |
|---|---|---|---|---|---|---|---|
| MobileBERT w/o OPT | 25.3M | 1.8× | 80.2 | 84.4 | 91.5 | **92.5** | 87.0 |
| L12-H384-E128 MiniLM | 25.3M | 2.7× | **80.6** | **85.2** | **91.7** | 92.1 | **89.5** |

Table 1: Results of 12-layer MiniLM with 384 hidden size and 128 embedding size.

**Visual analysis of attention weight transfer and Why distilling the last layer works better?**  We visualize the attention distributions of the teacher ($BERT_{BASE}$) and the 6x384 students using last layer and layer-wise distillation. We find that attention distributions for each layer of the layer-wise distilled student are very similar to its corresponding layers of the teacher. For the last-layer distilled student, attention distributions of the last layer mimic its teacher's last layer very well, while the bottom four layers are more similar to the teacher's bottom four layers. The fifth student layer is similar to teacher's middle or top layers. Last layer distillation also learns features of teacher's lower layers. Moreover, layer-wise transfer sets a tight restriction for each student layer. Transferring the last layer gives more flexibility for the bottom layers to learn the knowledge. Experiments (Table 8 of supplementary material) also demonstrate that relaxing restrictions of layer mapping improves performance. We will add the visual analysis in the revised paper.

# 2 Response to Reviewer #1

**Inconsistent results with TinyBERT paper.**  We focus on task-agnostic compression of pre-trained Transformers. As stated in the caption of Table 2, we compare task-agnostic distilled models without task-specific distillation (TD) and data augmentation (DA). Besides task-agnostic distillation, TinyBERT further uses TD and DA to achieve improvements for specific tasks, and reports the overall results. For a fair comparison, we fine-tune the latest version of their public task-agnostic model (using the same fine-tuning code and range of hyper-parameters) to report the results.

**Using $BERT_{LARGE}$ as teacher.**  Thanks for the suggestion. We are exploring it and will add the results in the future.

# 3 Response to Reviewer #2

**Which TinyBERT model we used?**  Besides the model with 14.5M parameters, TinyBERT also reports the results of 6x768 (6 layers, 768 hidden size, 66M parameters) model in Table 10 and 11 of their paper, and releases both the two models. Since most of previous works, such as DistilBERT and BERT-PKD, distill $BERT_{BASE}$ into a 6x768 student, we also adopt the same setting and compare with 6x768 TinyBERT.

**Analysis to what extent representations of the lower-level layers differ between the student and teacher model.**  Thanks for the suggestion. As stated above, visual analysis suggests that attention distributions of student's bottom four layers are similar to teacher's bottom four layers. We will try the methods you suggest and add the discussion.

# 4 Response to Reviewer #3

**Why not query-query or key-key relation but value-value relation?**  We have tried to add query-query or key-key relation, which improves the performance if we do not transfer attention distributions. But if we introduce attention distributions, it will not bring improvements. Attention distributions capture the relation between queries and keys. Knowledge of queries and keys can be transferred via distilling attention distributions. Transferring query or key relation has a similar effect as transferring attention distributions. Besides queries and keys, the remaining important vectors in self-attention module are values, so we introduce the value relation to achieve a deeper mimicry.

# 5 Response to Reviewer #4

**Why scaled dot-product between values performs better?**  1) Since self-attention module is vital in Transformer (attention is all you need), the key idea of our method is to deeply mimic the self-attention. Queries, keys, and values are the most basic and important vectors in self-attention, so we transfer attention distributions (relation between queries and keys) and value relation to achieve a deeper mimicry. 2) Using dot-product converts different dimensional vectors into relation matrices with the same size. Compared with transferring value vectors, it avoids introducing additional parameters (randomly initialized) to transform student vectors into the same size as its teacher. The additional transformation transforms vectors into another vector space and restricts teacher from directly transferring knowledge.

**How the student is initialized?**  Our students are randomly initialized. We will make it clear in the revised paper.

**Analysis of value relation.**  Thanks for the suggestion. We will add the analysis in the revised paper.

[Meta-Review · NeurIPS 2020]

This is another paper in a recent sequence of 'distilling contextualized word embeddings' papers where the primary innovation is that they only distill the final layer of the transformer -- also introducing a 'teacher assistant' mechanism (as introduced in previous work) to improve performance of the final student model. The result is simpler than competing work while performing better over a relatively extensive set of experiments (i.e., GLUE), especially if including the supplementary material. The reviews were positive to begin with and concerns were addressed during rebuttal -- thus, I recommend accepting for publication.